# Comparison of metabolic states using genome-scale metabolic models

**Chaitra Sarathy**[1]*, **Marian Breuer**[1], **Martina Kutmon**[1,2], **Michiel E. Adriaens**[1], **Chris T. Evelo**[1,2], **Ilja C. W. Arts**[1]

**1** Maastricht Centre for Systems Biology (MaCSBio), Maastricht University, Maastricht, The Netherlands, **2** Department of Bioinformatics - BiGCaT, School of Nutrition and Translational Research in Metabolism (NUTRIM), Maastricht University, Maastricht, The Netherlands

* chaitra.sarathy@maastrichtuniversity.nl

**Data Availability Statement:** All data and source code are available at https://doi.org/10.5281/

## Abstract

Genome-scale metabolic models (GEMs) are comprehensive knowledge bases of cellular metabolism and serve as mathematical tools for studying biological phenotypes and metabolic states or conditions in various organisms and cell types. Given the sheer size and complexity of human metabolism, selecting parameters for existing analysis methods such as metabolic objective functions and model constraints is not straightforward in human GEMs. In particular, comparing several conditions in large GEMs to identify condition- or disease-specific metabolic features is challenging. In this study, we showcase a scalable, model-driven approach for an in-depth investigation and comparison of metabolic states in large GEMs which enables identifying the underlying functional differences. Using a combination of flux space sampling and network analysis, our approach enables extraction and visualisation of metabolically distinct network modules. Importantly, it does not rely on known or assumed objective functions. We apply this novel approach to extract the biochemical differences in adipocytes arising due to unlimited vs blocked uptake of branched-chain amino acids (BCAAs, considered as biomarkers in obesity) using a human adipocyte GEM (iAdipocytes1809). The biological significance of our approach is corroborated by literature reports confirming our identified metabolic processes (TCA cycle and Fatty acid metabolism) to be functionally related to BCAA metabolism. Additionally, our analysis predicts a specific altered uptake and secretion profile indicating a compensation for the unavailability of BCAAs. Taken together, our approach facilitates determining functional differences between any metabolic conditions of interest by offering a versatile platform for analysing and comparing flux spaces of large metabolic networks.

## Author summary

Cellular metabolism is a highly complex and interconnected system. As many lifestyle diseases in humans have a strong metabolic component, it is important to understand metabolic differences between healthy and diseased states. In systems biology, metabolic behaviours are investigated using genome-scale metabolic models. In addition to the

zenodo.5495880 and github repository https://github.com/chaitrasarathy/ComMet.

**Funding:** This research has been made possible with the support of the Dutch Province of Limburg, The Netherlands. The funders had no role in study design, data collection and analysis, decision to publish, or preparation of the manuscript.

**Competing interests:** The authors have declared that no competing interests exist.

sheer size and complexity of the genome-scale metabolic models of human systems, using existing analysis methods is challenging and the parameter selection is not straightforward. Therefore, novel methodological frameworks are necessary for analysing metabolic conditions despite the challenges posed by human models. Particularly, an ongoing challenge has been that of comparing several phenotypes for identifying condition- or disease-specific metabolic signatures. We address this significant challenge by developing a scalable and model-driven approach, ComMet (Comparison of Metabolic states). ComMet enables an in-depth investigation and comparison of metabolic phenotypes in large models while also identifying the underlying functional differences. Novel hypotheses can be generated using ComMet for not only understanding known metabolic phenotypes better but also for guiding the design of new experiments to validate the processes predicted by ComMet.

## Introduction

Metabolism plays a central role in maintaining cell functionality as it provides the energy and building blocks for cellular growth. In humans, metabolic dysfunction is associated with a wide range of clinical conditions including obesity, diabetes, neurodegenerative diseases, cancer and inborn errors of metabolism [1–3]. Therefore, systems-level understanding of human metabolism is pivotal to comprehending phenotypic changes (in both normal and diseased states) and to develop prevention and treatment strategies.

Advancements in experimental and computational techniques have enabled the construction of genome-scale metabolic models (GEMs) in the last two decades. GEMs are mathematical formulations of the complete set of metabolic reactions taking place in a cell, tissue, organ or organism [4]. GEMs contain extensive descriptions of molecular relationships between genes, reactions and metabolites. These comprehensive knowledgebases enable prediction of reaction fluxes under varying environmental conditions thus contributing to systems-level understanding of metabolism. GEMs have facilitated investigating various metabolic dysfunctions in cancer [5–8], obesity [9] and non-alcoholic fatty liver disease [10, 11]. Despite successful applications and advances in algorithms [12], conducting studies with human GEMs still requires certain assumptions and/or prerequisites which are detailed below.

Flux Balance Analysis (FBA) [13], Elementary Flux Modes (EFM) analysis [14] and Flux Space Sampling (or Sampling) [15] are frequently used methods for analysing GEMs. Given the cellular uptake rates of metabolites, FBA optimises an assumed objective (such as biomass production) and estimates flux values for all reactions. However, the accuracy of FBA estimates predominantly depends on two factors: (a) assumed objective and (b) precise description of media or nutrient levels. Due to the complex nature of human cellular metabolism, selecting the objective function is not as straightforward as biomass production and requires careful consideration of the underlying physiology. In addition, the absence of accurate public data from human cell line studies (describing uptake or release rates of plasma metabolites) limits the applicability of FBA. Alternative to identifying a single optimal flux distribution, EFM analysis and Sampling characterise all possible flux states in the metabolic network. EFMs are non-decomposable steady-state pathways through a metabolic network. Owing to the combinatorial explosion in the number of EFMs, identification of the complete set of EFMs is computationally demanding [16], making it unsuitable for large GEMs. In addition, estimating the likelihood of observing an EFM in a given phenotype is difficult [17] which further limits their practical applicability on human GEMs. Sampling, on the other hand, provides a realistic

alternative to EFMs in exploring the properties of network states. Sampling identifies the feasible range as well as a probability distribution for every reaction flux in the model by generating uniformly distributed random points in the flux space (a geometrical polytope containing the set of feasible metabolic states) [15, 18]. Most importantly, unlike FBA, Sampling does not require the specification of an objective function. These methodological differences offer great benefits in using Sampling to assess metabolic differences in various physiological states. Due to the computationally intensive nature of the currently available algorithms for sampling human GEMs, we sought to develop a computationally practical method to analyse and compare metabolic states in large GEMs based on feasible flux space analysis as it does not require aforementioned assumptions.

In this study, we have developed a method for comprehensive analysis and comparison of large metabolic flux spaces called ComMet (**Com**parison of **Met**abolic states). ComMet provides a scalable and model-dependent framework that is computationally feasible and independent of objective specification. The functionalities available in ComMet allow in-depth characterisation of flux states achievable by GEMs followed by identification of metabolic differences between several conditions of interest. Characterisation of metabolic flux spaces involves identifying the key players and predicting flux statistics of the reactions active in that metabolic state. This identification is achieved in ComMet by building upon two existing approaches. First, the iterative algorithm developed by Braunstein et al., [19] gives an analytical approximation of the probability distribution of fluxes instead of describing this probability distribution through sampling random points within the flux space. The flux predictions obtained through their approach are as accurate as conventional Sampling algorithms and involve very little processing times. The computational efficiency makes this approach suitable even for large GEMs which was the focus of the present study. This approach by Braunstein et al., [19] will hereafter be referred to as "analytical approximation of fluxes" in this paper. Second, in an earlier study Barrett et al. [20] demonstrated that applying Principal Component Analysis (PCA) on a sampled flux space decomposes the flux states into biochemically interpretable reaction sets whose flux variability accounts for the variation in the entire flux space. Such a transformation extracts what are called "modules" (or biochemical features characterising a given state) based on network-wide flux interactions and provides useful insights into the underlying physiology. We build upon these two approaches following which we compare metabolic states and extract biochemical features that distinguish the different states. We first characterise the flux spaces of large GEMs by approximating the probability distribution of fluxes using [19]. Then, we perform PCA-based decomposition of flux space (adapted from [20]) which provides a basis for the subsequent comparison of flux spaces.

The novelty of ComMet lies in its ability to investigate differences between various metabolic states (for example, presence or absence of obesity). Metabolic features distinguishing the different conditions are extracted through rigorous optimisation of comparative strategies. The resulting distinctions are subsequently visualised in three network modes: reaction map, metabolic map and single module view. We demonstrate the applicability of ComMet using a large GEM, the metabolic reconstruction of human adipocyte, iAdipocytes1809 [9] as an example. This model serves the dual purpose of demonstrating ComMet and the potential biological relevance of the resulting predictions. We highlight the differences in the flux space of the adipocyte model arising due to presence or absence of branched-chain amino acids (BCAAs: leucine, valine and isoleucine). Elevated BCAAs are considered as strong biomarkers for obesity and diabetes [21, 22]. Although a mechanistic explanation for the observed increased levels is currently unavailable, impaired BCAA catabolism in adipocytes has been hypothesised as a contributing factor [23]. By extracting biochemically interpretable adipocyte modules, ComMet was able to provide additional insights into adipocyte-specific BCAA

metabolism. We validated predictions from ComMet by identifying molecular processes that were functionally related to BCAA metabolism. Apart from this, we also highlight the utility of ComMet as a tool for generating hypotheses that could potentially be tested in a laboratory setting.

## Results

First, this section will briefly explain the workflow underlying ComMet which is followed by the modules identified by ComMet from the adipocyte model (iAdipocytes1809 [9]). To demonstrate the value of ComMet for distinguishing between metabolic conditions, two metabolic states of an adipocyte were simulated: an unconstrained substrate uptake and blocked uptake of BCAAs. The simulated scenario demonstrates a proof of principle of ComMet's ability to investigate metabolic differences in various metabolic states.

### Development of the ComMet workflow

Starting with a GEM, ComMet describes an eight-step pipeline to analyse and compare metabolic flux spaces (Fig 1). The first step involved specification of constraints necessary for studying metabolic states of interest (Fig 1A). To identify the differences between unconstrained and blocked uptake of BCAAs, two metabolic conditions were simulated: (i) Unconstrained substrate uptake (Fig 1A, green: where all the exchange metabolites, including the uptake of BCAAs, were kept unlimited) and (ii) Constrained substrate uptake, (Fig 1A, purple: where only the uptake of leucine, valine and isoleucine were limited to zero). Specifying constraints for the metabolic states under study resulted in two condition-specific flux spaces. Each of them were preprocessed to remove any blocked reactions (Fig 1B). The preprocessed condition-specific flux spaces were decomposed into modules (sets of reactions having key contribution to the flux space) in the following manner. Analytical approximation of fluxes was carried out in both conditions (Fig 1C) using the algorithm developed by Braunstein et al [19]. Next, Principal Component Analysis (Fig 1D) and basis rotation (Fig 1E) were applied to each flux space. The covariance matrix resulting from the analytical approximation of fluxes was used for the PCA-based decomposition of the flux spaces (adapted from [20]). Such a decomposition determined the principal components (PCs or flux vectors) explaining the variation within each flux space. Subsequently, our analysis followed two separate directions to (i) identify condition-specific modules and (ii) compare metabolic conditions.

Essentially, the condition-specific modules contained sets of reactions whose fluxes contributed substantially in determining the underlying metabolic state. A module was extracted from each rotated PC which contained the most significant reactions within that vector (Fig 1G). The collection of modules from an individual flux space formed the "global modules" of that metabolic condition. To facilitate interpretation of modules and to identify the interplay between individual modules, the global modules were then visualised as reaction networks (Fig 1H). For comparing the two simulated metabolic conditions, the rotated PCs obtained previously were then subjected to Independent Component Analysis (ICA) which revealed vectors showing noticeable differences between the two conditions (Fig 1F). Finally, modules were extracted from these distinct flux vectors containing metabolic differences (Fig 1G) and were also visualised in three different forms: as a combined reaction map or network, metabolic map of subsystems of interest and single module views (Fig 1H).

### Analytical approximation of fluxes

During preprocessing, blocked reactions were defined as the reactions that are incapable of carrying any flux under the imposed conditions. The preprocessing step removed 2,043 and

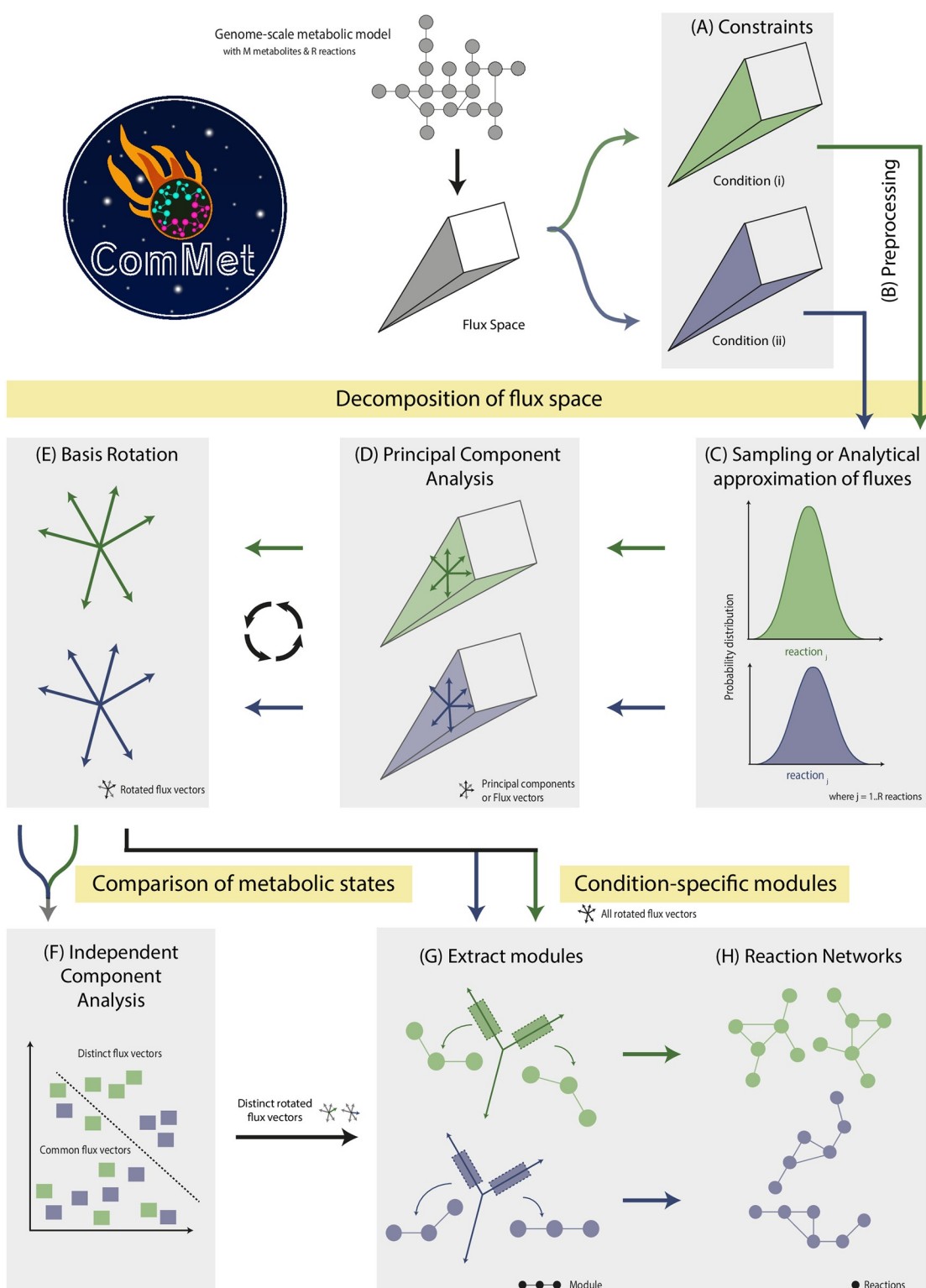

**Fig 1. Overview of the eight-step pipeline in ComMet to compare various metabolic states in genome-scale metabolic models (GEMs).** ComMet focuses on the flux space of an input GEM with M metabolites and R reactions. Once the reaction flux constraints are specified, condition-specific flux spaces are generated ((**A**) green and purple). Both flux spaces are then preprocessed (**B**) and decomposed into modules through Sampling or analytical approximation of fluxes (**C**) followed by Principal Component Analysis (**D**) and basis rotation (**E**). The decomposed flux spaces can be studied individually or compared using Independent Component Analysis (**F**) by extracting modules from all or distinct flux vectors respectively. Once the modules are extracted, they are visualised (**G-I**) in three forms: as a reaction map, metabolic map and as individual modules.

2,046 blocked reactions in the unconstrained and constrained models respectively, thereby retaining 4,067 and 4,064 reactions. These reactions formed a 1608-dimensional flux space. Subsequently, the algorithm by Braunstein et al., [19] allowed exploring the probability distributions of reaction fluxes in both conditions. Mean and standard deviation of flux distributions were obtained for every reaction. A histogram analysis was first performed to broadly understand the impact of imposing constraints (S1 Fig). The histograms of means from both conditions (S1(A) and S1(B) Fig) were unimodal and centred around 1 mmol gDW$^{-1}$ h$^{-1}$ (millimoles per gram dry weight per hour, the unit of flux used in GEMs). Both graphs were roughly symmetric with long tails extending till ± 1000 mmol gDW$^{-1}$ h$^{-1}$. The histogram of standard deviations (S1(C) and S1(D) Fig), on the other hand, were bimodal with a very large peak around 10 mmol gDW$^{-1}$ h$^{-1}$ and a smaller peak at 550 mmol gDW$^{-1}$ h$^{-1}$. The shape and spread of the corresponding histograms (S1(A) vs S1(B) and S1(C) vs S1(D) Fig) were notably similar between conditions. However, a closer inspection revealed differences.

In order to quantify the flux differences arising due to constraint imposition, next, a reaction-wise comparison of flux statistics was carried out. Fig 2 shows that a majority of the means and standard deviations lie on or close to, the identity line, suggesting a strong similarity in fluxes between the simulations. Deviations from the identity line seemed to indicate that the flux distributions of only a few reactions were visibly affected by limiting BCAA uptake. However, the mean fluxes were identical in only 308 reactions and the difference in means were between 0.01–10 mmol gDW$^{-1}$ h$^{-1}$ for 3,311 reactions. As detailed in S1 Table, about 267 reactions showed a change in mean flux between 10–700 mmol gDW$^{-1}$ h$^{-1}$. It is important to note that, as expected, the mean fluxes of several reactions involved in BCAA metabolism reduced to almost zero upon constraining BCAA uptake (black rectangle in Fig 2).

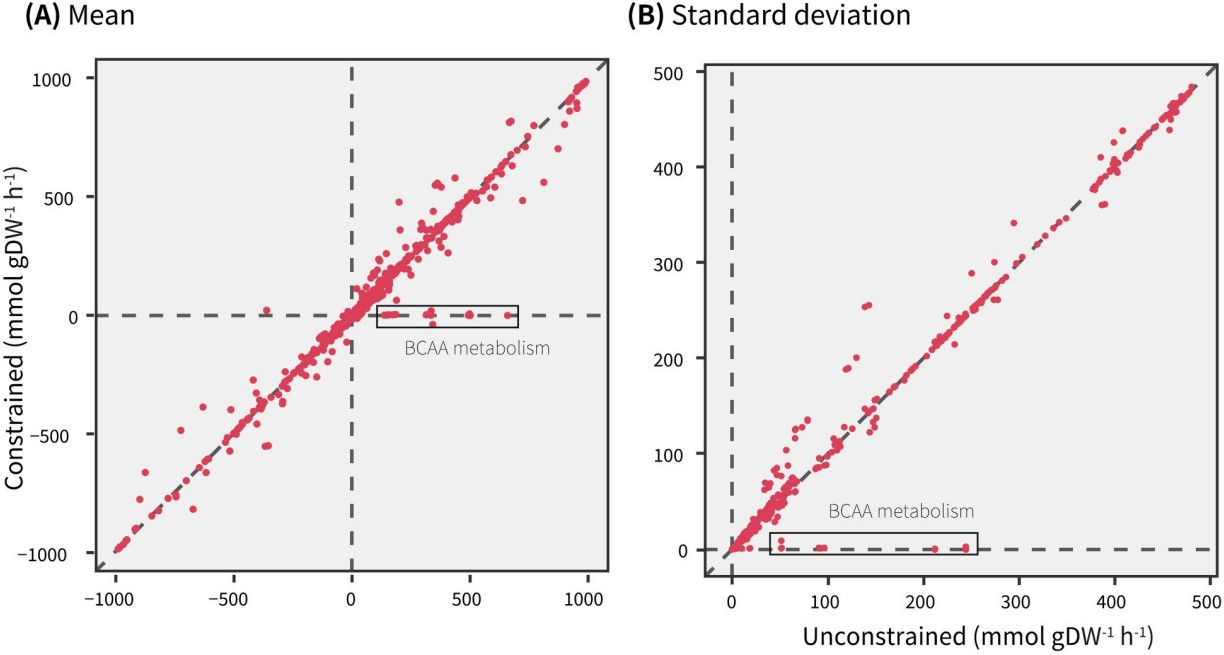

**Fig 2. Reaction-wise comparison of (A) means and (B) standard deviations between the unconstrained and constrained simulations.** The reactions involved in BCAA metabolism are highlighted in black rectangle.

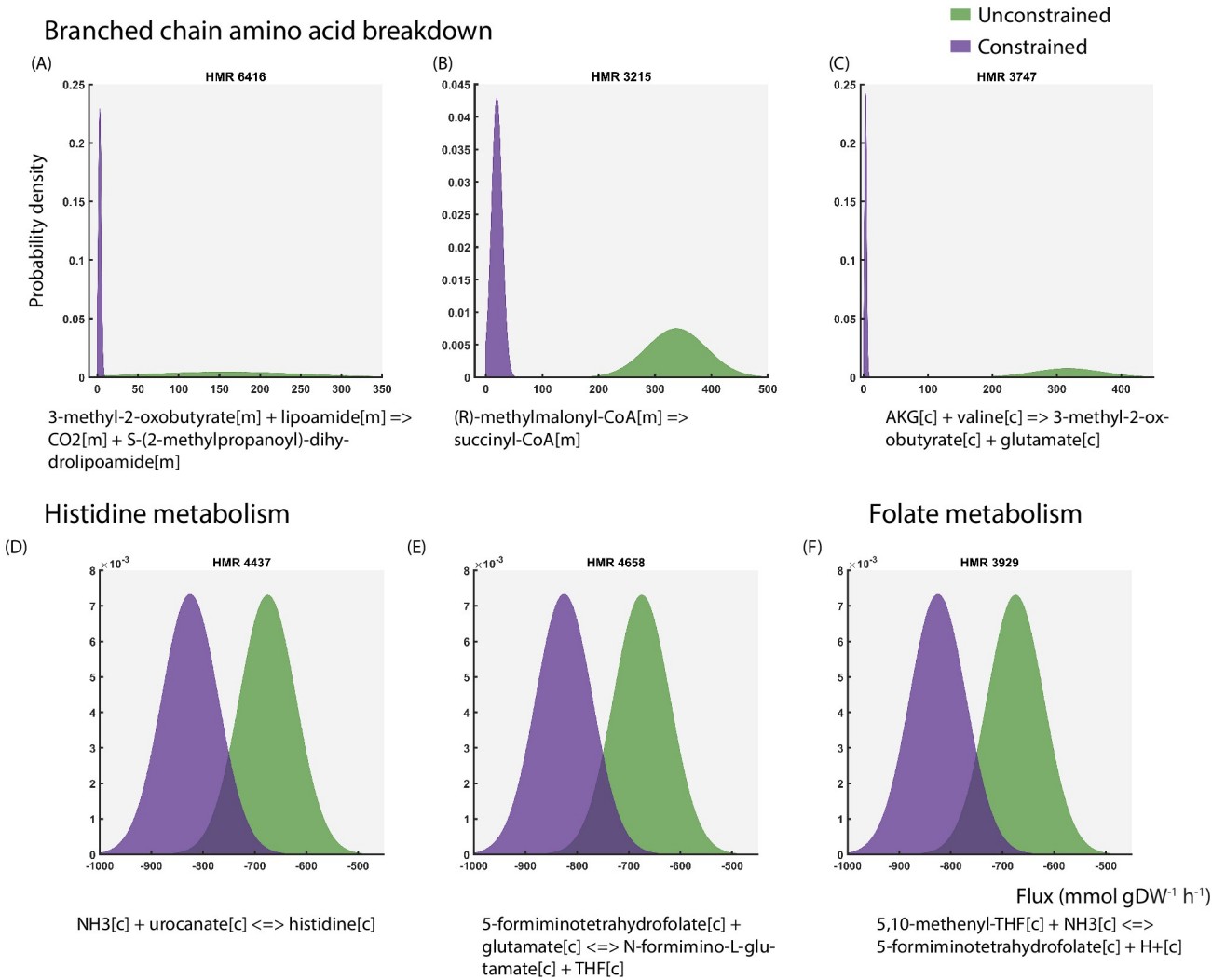

**Fig 3. Flux distributions of 6 out of 139 reactions having significant differences in flux statistics ($p < 0.05$) between the unconstrained and constrained simulations.** The reaction IDs and chemical equations have been shown above and below the plots respectively.

## Reactions with significantly different flux statistics

In order to evaluate the statistical significance of the flux differences, z-scores were computed using the means and standard deviations of reaction fluxes from both the conditions. Introduction of the constraint significantly affected the probability distributions through 139 reactions in the network ($p < 0.05$). The changes in flux distribution of all these reactions are shown using density plots (Fig 3, S2 Fig and S2 Table). The shape of each distribution gives information about the sensitivity of the solution space to the applied constraint. As evident from Fig 3A–3C and S2(A)–S2(I) Fig, in the constrained simulation, the flux distributions shifted to the left and spanned across a narrow range compared to a broad range in their unconstrained counterpart. This shift implied a reduction in the fluxes and as expected, this pattern was observed in several reactions in BCAA breakdown (Fig 3A–3C and S2A–S2F Fig). Additionally, the fluxes through the mitochondrial transport of intermediate metabolites involved in this pathway (methylmalonate and 3-methyl-2-oxobutyrate) also diminished (S2E

and S2F Fig). Application of the constraint also resulted in marginal release of BCAAs the potential sources of which includes breakdown of lipoproteins that is taken up from the *in silico* media.

On the contrary, the flux distribution through ammonia exchange not only changed in shape, but also reversed in reaction directionality. Under unlimited substrate uptake, our simulations indicated that ammonia is predominantly taken up whereas ammonia is predominantly released on limiting BCAA uptake (S2 Table). Interestingly, in the constrained simulation, the flux distribution through the reactions HMR_4437, HMR_4658 and HMR_3929 (Fig 3D–3F) retained the same shape and range as the unconstrained curve, but is shifted to the left on the negative axis. This indicated an increase in fluxes and was observed in the reactions releasing ammonia through folate and histidine metabolism, which could potentially contribute to the excess ammonia release.

Overall, the results of analytical approximation of fluxes seem to suggest that the restriction of merely three metabolite uptakes can affect the overall behaviour of the network, even in pathways whose connection to these metabolites is not obvious. However, this test inspected only the differences in individual reaction fluxes caused by the introduced metabolic perturbation without considering any interactions between reactions. Therefore, we followed a PCA-based approach to identify sets of interacting reactions contributing to the underlying metabolic states.

## Principal components

During analytical approximation of fluxes, a covariance matrix was also computed along with descriptive statistics of individual reaction flux distributions. The rows and columns of the matrix were equal to the number of reactions (4,067x4,067 in the unconstrained and 4,064x4,064 in the constrained simulations). Essentially, every column was a vector of covariance between the flux distributions of one reaction and all other reactions. Performing PCA on this matrix resulted in PCs explaining variation within each flux space. Each PC was a flux vector containing different values of loadings for all reactions. The graph in Fig 4 reports the percentage of cumulative variance explained by the PCs in both flux spaces. 99.9% of the variation in the metabolic flux spaces were explained by 519 and 515 PCs in the unconstrained and the constrained conditions respectively. The inset in Fig 4 zooms into the first 10 PCs and shows that the highest absolute variance is: 3.03% in the unconstrained and 0.86% in the constrained flux space. Individually, each of the remaining components explained less than 1% of the variation. Nonetheless, these results clearly demonstrate a considerable dimensionality reduction from 1,608 to approximately 500 dimensions in both flux spaces. This reduction also implies that the metabolic state of the adipocyte network can be largely set by regulating these 519 and 515 PCs respectively in the unconstrained and constrained flux spaces.

## Basis rotation

Next, applying a basis rotation on PCs allowed gaining a biochemically meaningful interpretation of the flux vectors by condensing the loadings of over 4,000 reactions within each PC into a few high-loading reactions. The reactions whose loadings were at least half of the largest absolute loading value in a given flux vector were considered to be part of the module. In other words, the variance explained by a component was driven by the high-loading reactions. Every module contained distinct sets of reactions which had either positive or negative loading values. We would like to reiterate that the collection of all the modules from an individual flux space formed the global modules of that metabolic condition. Biochemically, it means that the

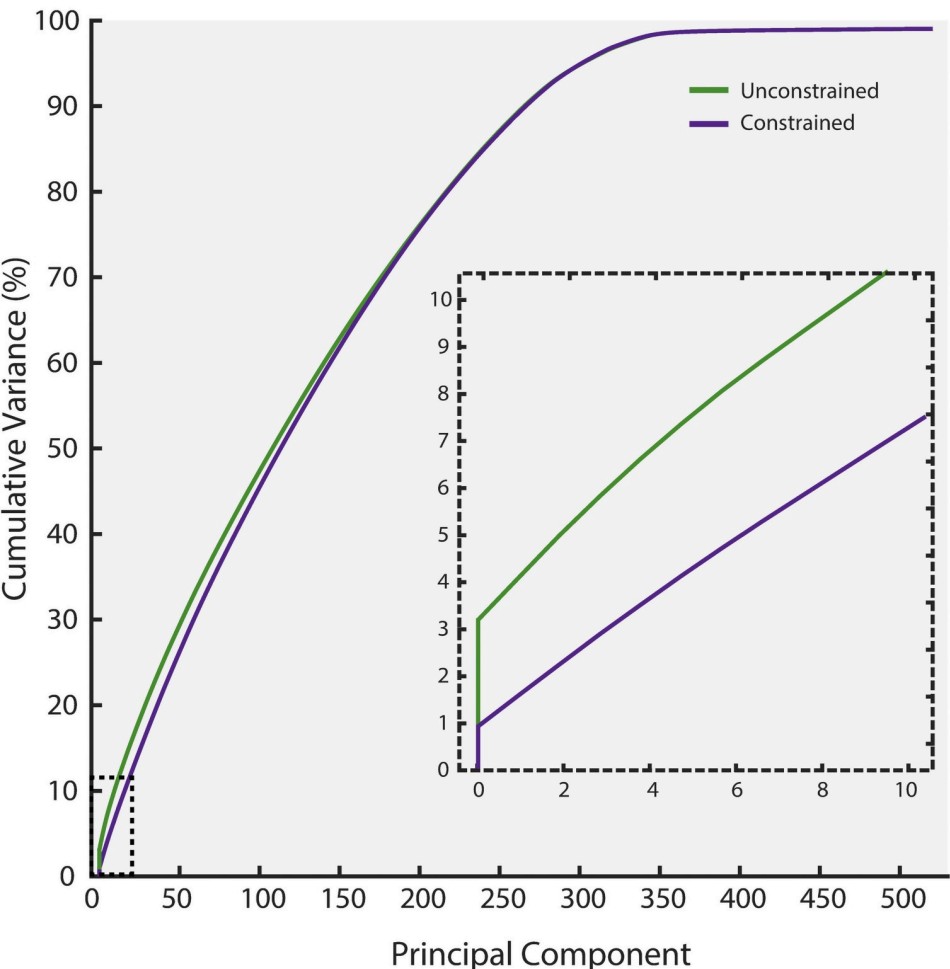

**Fig 4. Cumulative variance (%) explained by the principal components of flux spaces from both unconstrained and constrained simulations and the inset zooms into the first 10 principal components.**

reaction sets forming the global modules operate independently to maintain metabolic homeostasis in the underlying condition.

## Global modules in the adipocyte network: Unconstrained and constrained models

The global modules obtained from all the 519 modules in the unconstrained adipocyte network contained a total of 737 high-loading reactions. Although the reactions within a module were unique, some reactions were present in more than one module. In terms of size, the number of high-loading reactions in each module varied between 1 and 49. Constructing a reaction map from the reactions in the modules resulted in a reaction-reaction network (S3(A) Fig). Each node represents a reaction and the connections indicate shared metabolites (either a reactant or a product). This network contained 4,304 edges, one highly connected subset (564 reactions) and 75 unconnected reactions. The network structure of the reactions from the global modules of the constrained adipocyte (S3(B) Fig) was highly comparable to that from the unconstrained model (S3(A) Fig). The network in S3(B) Fig had slightly fewer reactions

(710) and few more edges (4,307). Just as seen in S3(A) Fig, this network also contained one highly connected subset (523 reactions) and 79 unconnected nodes. Both networks also contained small subsets of 2–5 reactions.

The connection between reactions and the modules was explored further by associating each reaction with its involved module(s). As highlighted in S3 Fig, the color of the nodes were mapped to the number of involved modules. This interlinking revealed that 85–90% of the reactions were present in just one module. Only ~12 reactions were present in 3 to (maximum of) 5 modules in both the networks. The dominance of single module reactions implied that very few reactions had a significant loading in multiple modules and most reactions contributed to only one PC of the flux space. In a metabolic context, they were responsible for regulating only one aspect of the network behaviour. Examining the biochemical processes (or subsystems) revealed that nearly half of the reactions in each network were involved in extracellular and mitochondrial transport. These nodes were also responsible for the high network connectivity. A list of all the unique subsystems with the number of reactions in each is given in (S3 and S4 Tables). The NDEx links https://bit.ly/globalModulesUncon and https://bit.ly/globalModulesCon can be used to study these networks interactively.

## Identification of BCAA-specific modules

ICA originated in the field of signal processing for separating individual sources from a mixture of non-Gaussian signals. In our analysis, ICA enabled identification of the rotated PCs that were distinct between the conditions, thus, contributing to the metabolic differences between the simulations. ICA was performed on the combined set of 1,035 rotated PCs from both conditions (519 and 515). An optimisation was carried out to address the algorithmic stochasticity and to select the number of independent components (ICs or features to be estimated, N).

ICA identified 288 distinct features among which, 150 PCs were from unconstrained and 138 PCs from constrained simulations. Module extraction followed by construction of a combined reaction map resulted in a network of 203 reactions (Fig 5). Colouring the nodes of the network by condition showed that 52 reactions were unique to the unconstrained distinct modules (green nodes) whereas 15 reactions were present only in the distinct constrained modules (violet nodes). 136 reactions were part of the distinct modules from both conditions (orange nodes). Mapping the nodes or reactions to the associated subsystem (as per model annotation) and grouping the nodes by subsystem revealed the involvement of 33 subsystems affected by the introduced perturbation (S5 Table). Addition of first neighbouring reactions from the original adipocyte model (that were not part of distinct modules), shown as grey nodes, improved the connectivity between the subsystems. Furthermore, laying out functionally similar subsystems together showcased the process-level crosstalk across the network and thus, enabled obtaining a more complete picture of the intracellular metabolism. The NDEx link https://bit.ly/DistinctModules can be used to study this network interactively. The annotations of eighteen subsystem groups are shown in Fig 5, three of which are highlighted in blue underlay (left: TCA cycle, middle: BCAA metabolism and mitochondrial transport of intermediates and right: Glycerine, serine and threonine metabolism and Folate metabolism). The details of remaining affected subsystems can be found in S5 Table. As expected, both cytosolic and mitochondrial BCAA breakdowns were affected by zero BCAA uptake and were observed only in the unconstrained modules (Fig 5, middle underlay). The biochemical relationships between BCAA metabolism and other subsystems predicted in this study were checked for consistency with existing knowledge and are detailed below. These subsystems have also been visualised as metabolic maps in Fig 6.

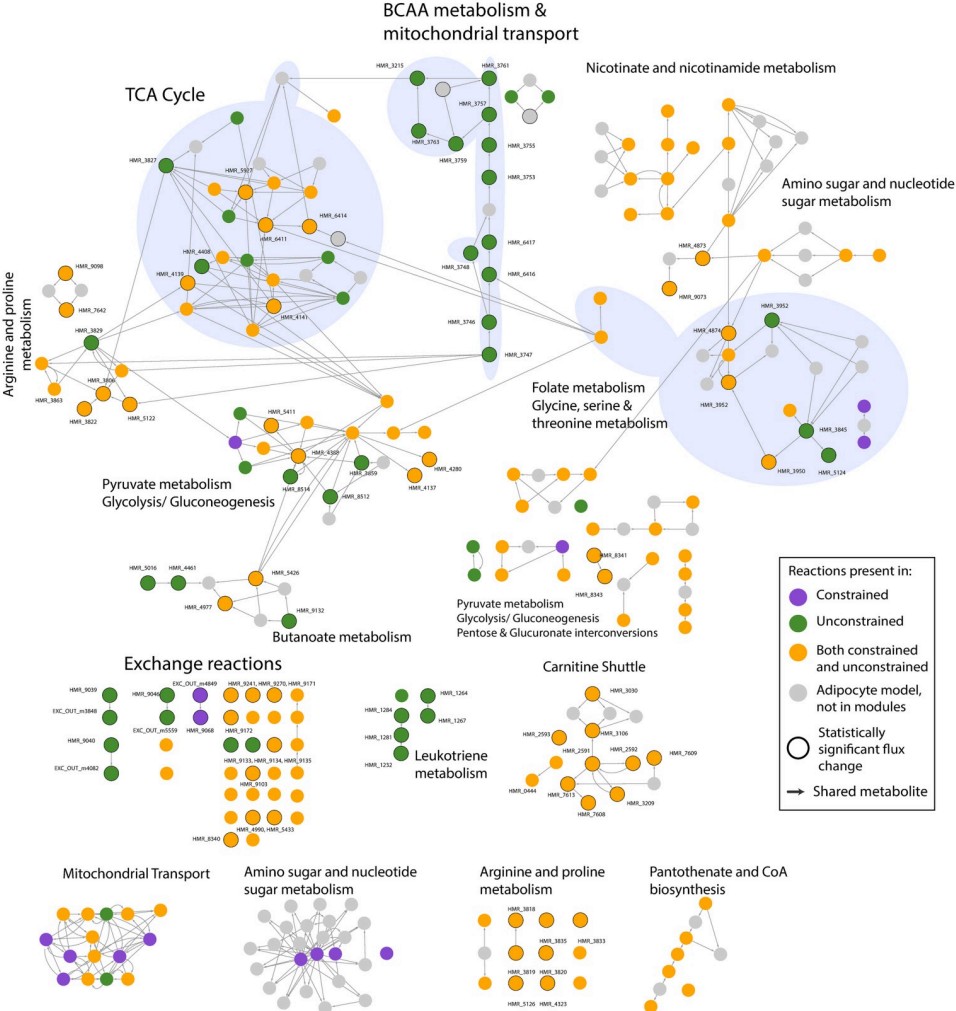

**Fig 5. Combined reaction map or network of modules extracted from the 288 PCs that were biochemically distinct between the unconstrained and constrained simulations.** Nodes represent reaction and colours indicate the condition: unconstrained (green), constrained (purple) and both (orange). Reactions are grouped by subsystems, some of which are highlighted in blue underlay (left: TCA cycle, middle: BCAA metabolism and mitochondrial transport of intermediates and right: Glycerine, serine and threonine metabolism and Folate metabolism). Grey nodes are the first neighbours of the reactions in distinct modules extracted from the adipocyte model. The reactions containing significantly changed fluxes ($p < 0.05$) are shown as bigger nodes with a black outline. The NDEx link https://bit.ly/DistinctModules can be used to study this network interactively.

**Comparison of ComMet predictions with existing knowledge.**

**BCKDH knockdown**. Knockdown of BCKDH (branched-chain alpha-ketoacid dehydrogenase, an enzyme in BCAA degradation) has been shown to reduce BCAA catabolic activity in several studies involving various biological systems [24]. The blocked BCAA uptake simulated here resulted in a similar observation. The distinct modules network contained the cytosolic and mitochondrial BCAA breakdown reactions and the transport of the intermediates between mitochondria and cytosol (Fig 5, middle underlay) only in the presence of BCAA uptake (Figs 5 and 6A).

**Fatty acid metabolism**. Through [13C] labelling studies, Green et al., [24] showed that BCAAs fuel TCA cycle and lipogenesis in adipocytes. The end products of mitochondrial BCAA catabolism in adipocytes include succinyl-CoA and propionyl-CoA which are then

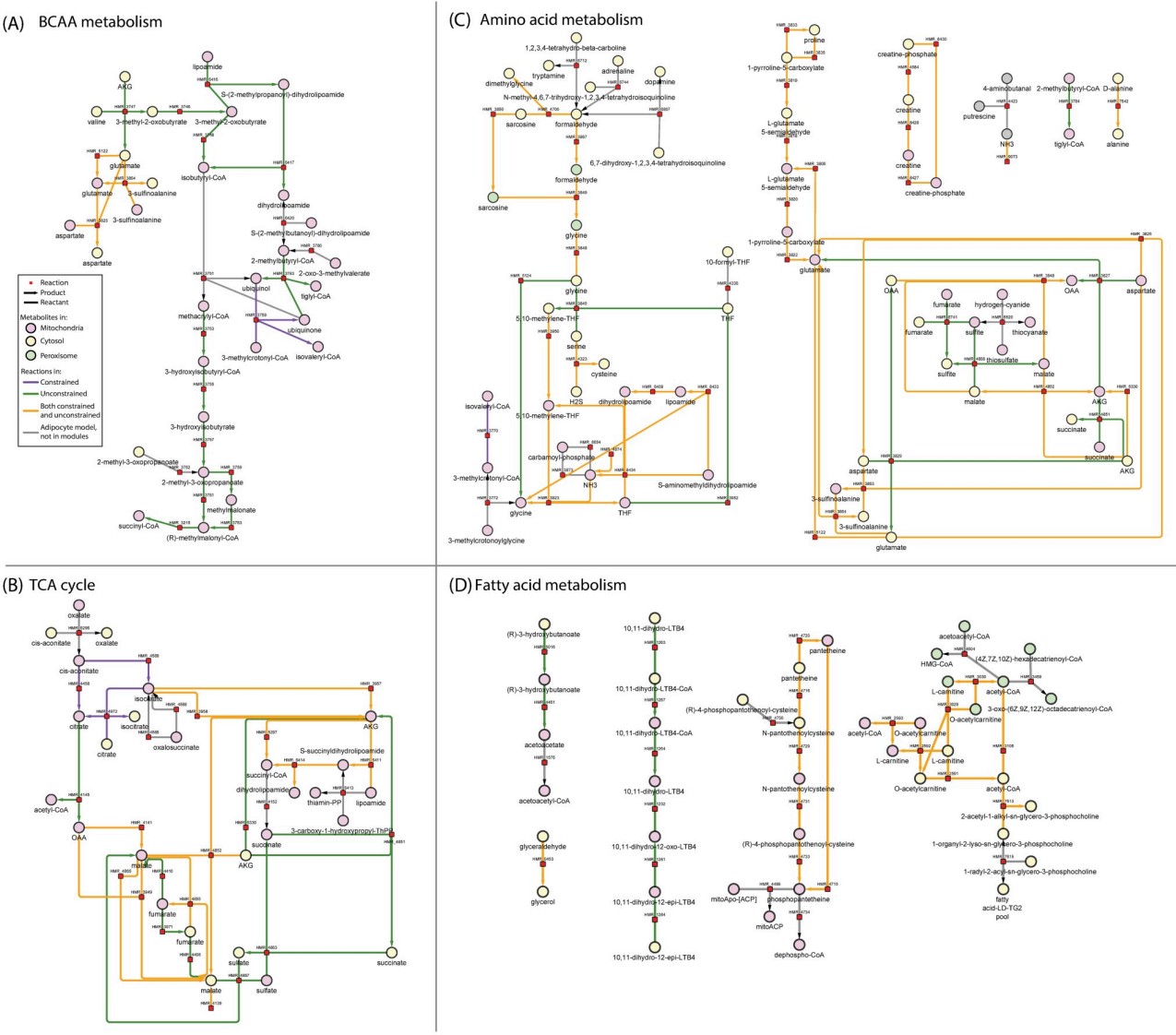

**Fig 6. Metabolic map of selected reactions from Fig 5 that were part of distinct modules.** Four subsystems are shown:(A) BCAA metabolism, (B) TCA cycle, (C) Amino acid metabolism and (D) Fatty acid metabolism. The nodes in this network represent metabolites (circles) and reactions (squares) while the edges correspond to reactants and products. Metabolite node colours indicate the intracellular location, mitochondria (pink), cytosol (cream) and endoplasmic reticulum (grey). The colour scheme of the edges follows that of the reactions in Fig 5.

metabolised via TCA cycle into lactate and acetyl-CoA. These metabolites are then channeled towards Fatty acid (FA) metabolism. Through our analysis, the metabolic map of BCAA metabolism (Fig 6A) clearly shows the detailed BCAA catabolic pathway leading to succinyl-CoA. The link between BCAA metabolism and TCA cycle (via succinyl-CoA) can also be observed in Fig 5 (connection between left and middle overlays). Therefore, the distinct modules resulting from our simulations are in line with these experimental observations.

Moreover, Fig 5 reveals the reactions downstream of TCA cycle, i.e., subsystems related to FA metabolism such as Carnitine shuttle, Leukotriene metabolism, Pantothenate metabolism, Beta oxidation, Glycerolipid metabolism, Glycerophospholipid metabolism and Butanoate metabolism (Fig 6D). Notably, Leukotriene metabolism was active only in the case of

unlimited BCAAs, indicating that these reactions are more significant when BCAAs are available to the cell. On the other hand, the mean reaction fluxes (S5 Table) showed an increase in Carnitine shuttle, Pantothenate metabolism, Glycerolipid metabolism, Glycerophospholipid metabolism and Beta oxidation upon limiting BCAA uptake. Our observation is in line with a recent study on rat adipocytes which reported an increase in palmitate oxidation (which occurs via beta oxidation) when external leucine levels reduce [25]. Thus, the increase in mean fluxes of FA metabolism, as shown by our analysis, could be a compensatory response to the decreased availability of acetyl-CoA from BCAA breakdown. Overall, the distinct modules suggest that FA metabolism is affected by BCAA availability in the adipocytes (Figs 5 and 6D).

**Carbon metabolism**. With respect to carbon metabolism, the distinct modules showed a reduced flux in pyruvate decarboxylation in the presence of BCAAs (HMR_4137, HMR_6410, HMR_6412). This observation is in line with the experiments on rat adipose tissue which demonstrated a decreased $CO_2$ release from glucose and pyruvate when leucine was provided [26].

As indicated by yellow nodes in Fig 5 (left underlay), the reactions from TCA cycle were present in both conditions. However, there were notable differences in the mean fluxes of reactions converting alpha-ketoglutarate to succinyl-CoA (S5 Table). In the absence of BCAAs, there was an increased flux in reactions breaking down alpha-ketoglutarate (HMR_6411, HMR_6414, HMR_5297, bigger nodes with black outline inside the left underlay in Fig 5) which were also identified as statistically significant ($p < 0.001$, S2 Table). Thus, our analysis predicts an increased flux towards succinyl-CoA with no BCAA uptake (Fig 6B). On examining the individual constituent module, these reactions were found to be part of the distinct module PC # 489 from the unconstrained model. This module contained the highest number of reactions and is visualised in Fig 7A. Two routes producing succinyl-CoA were present in this module in the presence of BCAAs. It contained the BCAA catabolic pathway resulting in succinyl-CoA and its production from alpha-ketoglutarate (HMR_5297) (Fig 7A). While the route from BCAA catabolic pathway was only active when BCAAs were available (green nodes, Fig 7A), the reaction HMR_5297 (yellow node, Fig 7A) was also found to be active in the absence of BCAAs (PC #379 constrained model, S5 Table). The presence of alternate routes towards succinyl-CoA indicates an anti-correlation between the two routes whose utility depends on the presence of BCAAs.

As mentioned above, the end products of BCAA metabolism have been shown to enter TCA cycle as succinyl-CoA [24]. The observed differences in mean fluxes along with observations from the module structure strongly suggest that in the absence of BCAAs, metabolic products of alternate sources (such as other amino acids) enter TCA cycle upstream of succinyl-CoA as alpha-ketoglutarate. Overall, the distinct modules demonstrate that the absence of BCAA uptake affects both TCA cycle and FA metabolism.

**Novel mechanisms predicted by ComMet.**

**Amino acid metabolism**. Subsystems metabolising several amino acids (phenylalanine, tyrosine, tryptophan, glycine, serine, threonine, arginine, proline, alanine, aspartate, glutamate, cysteine and methionine) can be found in Fig 5 and S5 Table. They are also visualised as a metabolic map in Fig 6C). Amino acids are metabolised by adipocytes and utilised for energy production through the TCA cycle with subsequent usage in fatty acid metabolism. The edges in the reaction map in Fig 5 show a clear connection between the subsystems metabolising amino acids and TCA cycle (for example, links between the left (TCA cycle) and right underlay (Glycine, serine and threonine metabolism) in Fig 5).

As indicated by purple edges in Fig 6C, the exchange of proline with the *in silico* media (HMR_9068) was exclusive to the modules from limited BCAA uptake. In addition, proline and arginine metabolism increased and alanine breakdown reduced when BCAAs were unavailable (S5 Table). The release of alanine (HMR_9098) and glutamate (HMR_9071) into

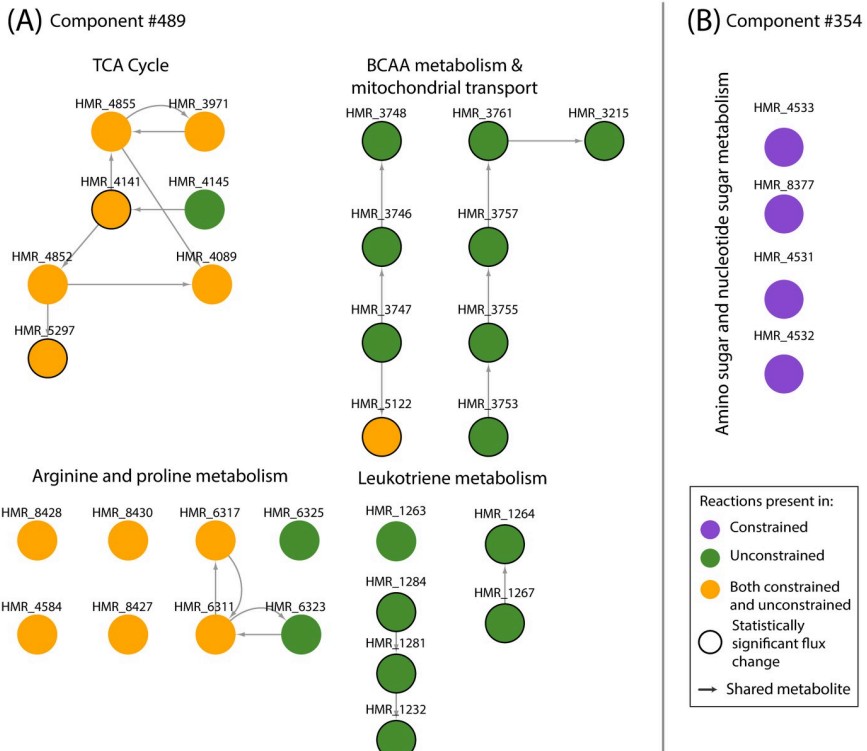

**Fig 7. Visualisation of the reaction sets in individual modules from PC (A) #489 from the unconstrained model and (B) #354 from the constrained model.**

the *in silico* media also increased when BCAAs were available (S5 Table). The cytosolic and mitochondrial breakdown of aspartate (HMR_3827 and HMR_3829) were exclusive to availability of BCAAs (Fig 6C). The changes in amino acid profile suggests that adipocytes could be compensating for the lack of BCAAs by increasing the catabolism of proline and arginine towards TCA cycle via alpha-ketogluterate. Our predictions on increased alanine and glutamate release in the unconstrained scenario are consistent with the experimental studies that demonstrated that the presence of leucine [26, 27] or valine [27] significantly increases the *in vitro* release of alanine [26, 27], glutamine and glutamate [26, 27] in rat epididymal fat pads.

Decreased glycine secretion was reported in differentiated mice adipocytes cultured on low BCAA levels [24]. In another study, Alves et al., [28] found that circulating glycine levels were increased in obese rodents and these levels decreased when restricting dietary BCAA intake. In our analysis, increased peroxisomal glycine formation/transport (HMR_3848, HMR_3849) along with increased mitochondrial glycine flux (HMR_3923) and cytosolic glycine breakdown (HMR_4700) was observed when BCAAs were unavailable (Figs 5 and 6C and S5 Table). Thus, the additional reactions metabolising glycine provide a possible explanation for glycine reduction in absence of BCAAs (right underlay in Fig 5).

**Metabolite exchanges with *in silico* media**. Several notable differences were observed between the simulations in the exchange profile of the adipocytes. The distinct modules indicate that release of BCAAs (EXC_OUT_m3848, EXC_OUT_m4082, EXC_OUT_m5559) were exclusive to the unconstrained simulation (green nodes under Exchange reactions in Fig 5) and ammonia exchange was present in both simulations (HMR_9073, Fig 5 and S5 Table). Intriguingly, under unlimited BCAA uptake, ammonia was taken up, whereas ammonia was

released on limiting BCAA uptake (as discussed under Analytical approximation of fluxes). In particular, the availabiltiy of BCAAs seemed to have affected the profile of ketone bodies (3-hydroxybutanoate and acetoacetate) and non-esterified fatty acids (NEFAs). In the absence of BCAAs, both uptake and release of NEFAs (HMR_9033 and HMR_9056) increased (S5 Table). Whereas, the uptake of 3-hydroxybutanoate (HMR_9134) and release of acetoacetate (HMR_9132) were exclusive to unconstrained modules (Fig 5 and S5 Table). Additionally, there were interesting observations in the profile of carbon sources and amino acids. The uptake of glucose (HMR_9034) and fructose (HMR_9139) were present in both simulations with marginal differences in their mean fluxes (S5 Table). On the other hand, the uptake of glycerol (HMR_9085)and xylitol (HMR_9139) increased while glucosamine (HMR_9168) uptake, L-arabinose uptake (HMR_9270) and L-arabitol release (HMR_9241) reduced when BCAAs were unavailable (Fig 5 and S5 Table). Furthermore, pyruvate uptake (HMR_9133) was exclusive to unconstrained modules and L-lactate release (HMR_9135) significantly reduced in the absence of BCAAs ($p < 0.001$, S2 Table, Fig 5 and S5 Table). As for the profile of amino acids, sarcosine uptake (HMR_9131) increased while the release of cysteine, glutamate, serine and D-alanine (HMR_9065, HMR_9071, HMR_9069 and HMR_9098 respectively) reduced upon blocking BCAA uptake. Taken together, the metabolites describing the changed exchange profile can be used to validate the predictions from our study.

## Distinct modules and flux statistics

The results from the BCAA-specific modules were compared with those from analytical approximation of fluxes alone. The 203 reactions from the distinct modules (Fig 8A) and the 139 reactions with significantly changed fluxes (Fig 8B) have been highlighted on the same plot that compared reaction-wise flux means (Fig 2A). Conversely, 83 out of these 139 reactions were also found in the distinct modules (Fig 5, bigger nodes with a black outline). As seen from Fig 8, the number of reactions identified in PCA-based analysis were substantially

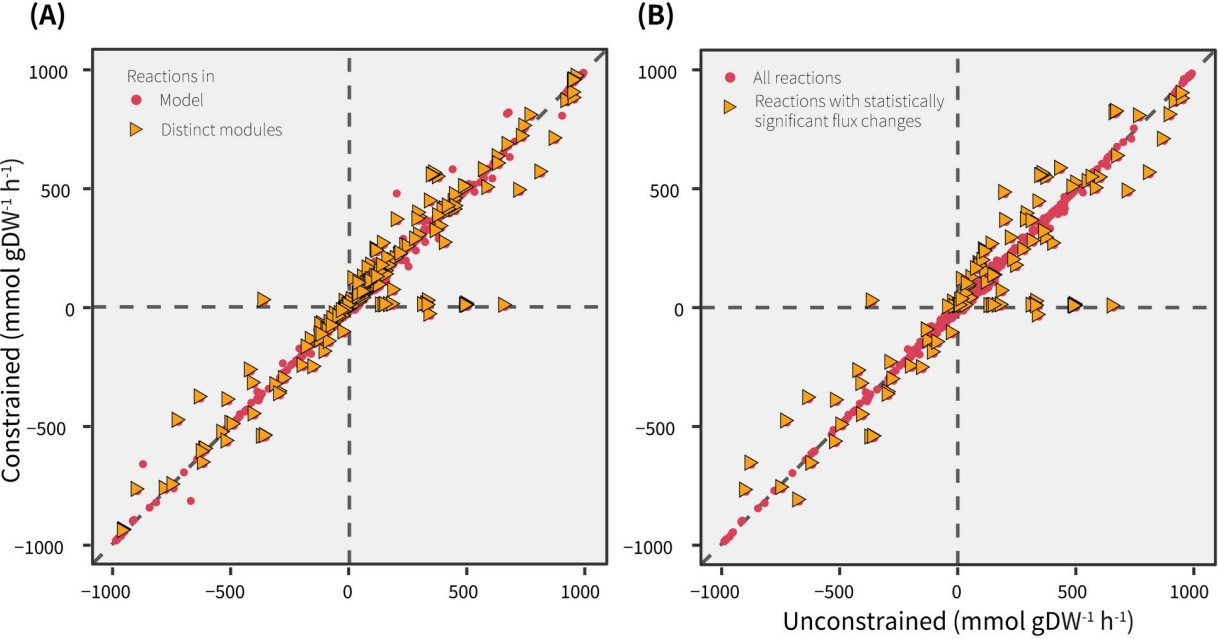

**Fig 8. Reaction-wise comparison of means between the unconstrained and constrained simulations (same as Fig 2A) highlighting (A) reactions from the distinct modules and (B) reactions with significantly changed fluxes in yellow triangles.**

higher than the reactions with significantly changed flux statistics. Although the mean fluxes of most reactions from distinct modules appear visually comparable in Fig 8, they showed statistically significant differences between the simulations S5 Table. Nonetheless, the distinct modules revealed biologically meaningful connections between BCAA breakdown and other subsystems (described above) that were not found from analytical approximation of fluxes alone.

### Distinct modules and subsystem annotations

To understand how distinct modules differed from subsystem annotations, the module structure was compared with subsystems defined in GEMs. Broadly, subsystems provide merely one form of annotating reactions in GEMs. They can be considered equivalent to pathways and are classified based on the type of biological macromolecule (for example, protein, fat, carbohydrate). As subsystems represent the biochemical processes in a cell, most of the subsystem annotations in GEMs remain the same across organisms and cell types. On the other hand, the reaction modules identified in this study (Fig 5) describe the most significant functional differences specifically between the simulated metabolic conditions. Moreover, the reactions from a module are extracted from one PC and can either belong to one subsystem or span across several subsystems. Fig 7 highlights two such single modules which have been extracted from the combined distinct module network (Fig 5). Fig 7A shows the module from component 489 (unconstrained model) which contains reactions from the subsystems TCA Cycle, BCAA metabolism, Mitochondrial Transport, Arginine and proline metabolism and Leukotriene metabolism. On the other hand Fig 7B shows the reactions extracted from component 354 (constrained model). Four reactions of the subsystem Amino sugar and nucleotide sugar metabolism (out of 28 reactions in this subsystem) are seen in this module. Overall, the modules describe cross-talk between various cellular processes and thus, are useful for inferring the network-wide biochemical changes between different metabolic conditions.

## Discussion

In this paper we present a novel method, ComMet, for in-depth characterisation and comparison of distinct metabolic states, which has remained a challenge for large metabolic networks. ComMet facilitates investigating several metabolic states through rigorous optimisation of strategies for comparing metabolic states followed by intuitive network visualisation. Using ComMet, biologically interpretable modules associated with different metabolic states of the adipocyte model (iAdipocytes1809) were extracted.

ComMet was first applied to explore adipocyte flux space and determine condition-specific modules. Principal components of the individual flux spaces uncovered that the metabolic state of an adipocyte network can be set by regulating ~500 dimensions. The corresponding modules spanned across a wide array of subsystems thereby controlling various aspects of adipocyte functionality. Using BCAAs as an example, ComMet was then used for identifying differences in adipocyte flux spaces arising from blocked uptake. Histograms of the flux statistics obtained from analytical approximation provided a broad sense of the flux ranges in each condition. The similarity in shape and spread of the corresponding histograms (S1(A) vs S1(B) and S3(C) vs S3(D) Figs) suggested a close correspondence in the overall flux profile between conditions. However, subsequent reaction-wise comparison followed by evaluation of statistical significance indicated otherwise. Identifying reactions with shifts in flux means and/or standard deviations provided a general outline of affected metabolic pathways. Since metabolic reactions seldom act independently, the downstream PCA-based analysis and meticulous comparison were vital in determining network-wide consequences of the introduced

perturbation. The observations from only comparing flux statistics failed to highlight metabolic pathways and connections that were indicated in distinct modules and these pathways were even experimentally shown to be affected in other studies. Most notably, the connections between BCAA metabolism and TCA cycle and the differences in metabolite exchanges, resulting from blocked BCAA uptake, were revealed in the distinct modules. Moreover, the changes in glycine breakdown profile predicted by the distinct modules, provide a possible explanation for the reduction in extracellular glycine levels in adipocytes upon limiting BCAA uptake. Many of the ComMet predictions align well with several experimental findings from mice and rat adipose tissues. These predictions serve as hypotheses to design experiments that can validate the predicted mechanisms on human cell lines.

## Innovations of the present study

The central conceptual advance of ComMet lies in its ability to compare multiple flux spaces, extract distinct metabolic features and visualise the extracted features (Fig 1).

By using the principal component based approach to decompose the flux space of the adipocyte model in two different metabolic states, we showed that biochemically interpretable modules can also be obtained in large-scale models (S5 Fig). Following this decomposition, the analysis was extended by introducing the central aspect of comparing decomposed flux spaces using ICA which was not trivial at all. The analysis presented here demonstrated an application of ICA in the context of flux spaces. ICA is a useful technique for source separation and has found successful uses in the field of image analysis. Here, ICA enabled identifying the flux vectors that are biochemically different in the compared metabolic states. Despite its stochasticity, rigorous ICA optimisation strategies were incorporated which strongly support reproducibility and ensure data-dependent parameter selection.

Moreover, three novel and reusable visualisation strategies were presented here which facilitated comprehensive understanding of the network behaviour under the imposed constraints. These strategies looked at the modules from three different angles, with each angle offering a unique perspective to interpreting the modules. The reaction maps or networks in Fig 5 combined the reactions from all the distinct modules into one network and showcased the reaction connectivity across the network. Additionally, grouping of these reactions based on subsystems provided an overview of the metabolic processes affected by the constraint introduction. The metabolic maps shown in Fig 6 zoomed into individual subsystems (selected from Fig 5) and revealed the underlying reactants and products. This representation offered a conventional pathway-like view and provided an indication of how the flow of metabolites in a subsystem may be affected by the constraint introduction. Fig 7 focused on a single module identified to be distinct between the two conditions in Fig 5. This view showed reaction connectivity within a module and indicated which reactions show correlated behaviour of different processes within that principal component. This allowed detailed examination of the reaction sets that govern specific biochemical aspects of the underlying metabolic state.

Overall, the innovative aspect of comparing large flux spaces and visualisation strategies developed here, enables investigating a wide range of metabolic conditions even in human-sized GEMs which are highly relevant in a biomedical context.

## Applications of ComMet

Extracting the biochemical differences from two simulated states of an adipocyte model opens avenues for extensive exploration of several metabolic conditions. Although ComMet was demonstrated here using a large GEM, it can also be used on smaller microbial models. For smaller GEMs, the underlying workflow may be adapted to obtain probability distributions of

reaction fluxes with either the analytical approximation scheme or with conventional Monte Carlo sampling as that would be computationally feasible for smaller GEMs. ComMet can not only be applied on diverse biological systems, ranging from microbe to human models, but it also allows analysing any metabolic states of interest, for example, comparing (a) metabolic shifts between healthy and cancer cells, or (b) metabolic capabilities of different members of gut microbiome, or (c) identifying metabolic differences between cell/tissue types, to name a few. Broadly, ComMet can be employed in two independent scenarios: (1) model-based hypothesis generation or (2) data-driven analysis. As a proof of principle, the first approach was demonstrated here on a human adipocyte GEM. The iAdipocytes1809 model was used as an example to not only demonstrate the new method but also to showcase the potential biological relevance of the resulting predictions. The simulated scenario (presence/absence of BCAAs in an adipocyte) is not meant to represent the 'true' physiology under obese/nonobese conditions. With a perturbation of merely three reactions, our approach identified several network-wide downstream effects and gave rise to several testable hypotheses. Such a model-based approach can also be extended on a much greater scale by blocking multiple (or combinations of) uptake metabolites. On the other hand, using experimental data in place of simulated states follows a data-driven approach. When available, omics data can be used to constrain flux spaces (Fig 1A) in place of simulated states to study physiologically accurate scenarios.

## ComMet and other approaches

In-depth characterisation and comparison of metabolic flux spaces is achieved in ComMet by building upon two existing approaches: PCA decomposition [20] and analytical approximation of flux space [19]. In their study, Barrett et al., [20] demonstrated a principal component-based decomposition to analyse the structure of a single flux space. ComMet adapts the PCA decomposition approach to obtain biochemically interpretable reaction sets of multiple flux spaces (i.e., two example metabolic states of an adipocyte) (Fig 1C–1E). Thus the decomposition of an individual flux space provided a basis for the analysis and subsequent comparison flux spaces. Notably, the analysis of large flux spaces was not addressed in the Barrett study [20] and their approach was demonstrated using a much smaller *E. coli* model.

Regarding the analytical approximation of flux space, Braunstein et al., [19] demonstrated a tremendous computational advantage of their Expectation Propagation (EP) algorithm over Monte Carlo sampling even for models with thousands of reactions. Moreover, it was also shown that the flux predictions obtained through their approach were as accurate as conventional Sampling algorithms. Therefore, the analytical approximation approach was chosen in this study to address the technical challenges arising from large-scale GEMs (Fig 1C).

The predictions resulting from ComMet are different from what can be obtained by approaches like FBA. ComMet and FBA are fundamentally different. They address two different research questions and have different technical requirements. Setting a metabolic objective and defining precise quantities of media metabolites are indispensable for a meaningful FBA simulation. For human GEMs, selecting an objective function is not as straightforward as biomass production and requires careful consideration of the underlying physiology. Due to the methodological differences, the two approaches find applications in very different scenarios. We would like to emphasise that ComMet is an approach that complements the existing methods for conducting studies with GEMs. ComMet is recommended for cellular systems, particularly in human tissue models, where assuming an objective is tricky and when accurate metabolite constraints are unavailable or difficult to obtain.

By showcasing the adipocyte modules that are affected by differences in BCAA uptake, we would like to emphasise the utility of ComMet as a tool for generating hypotheses which could be tested in a laboratory setting. Taken together, we demonstrate that ComMet is a powerful tool for holistic understanding of cellular physiology in several metabolic states.

## Materials and methods

The entire analysis presented in the current study was carried out in MATLAB R2017b [29] and all the networks were visualised using Cytoscape v3.7.2 [30]. The genome-scale metabolic reconstruction of human adipocyte, iAdipocytes1809 [9], was used. Model import and other model-related operations were carried out using the RAVEN toolbox [31]. The iAdipocytes1809 model contained 1,809 genes, 6,110 reactions and 4,361 metabolites. Reactions releasing leucine, isoleucine and valine from cytosol to extracellular space were added to the model. The entire analysis was carried out on a workstation running Windows 10 with E5–1650 6-core 3.5 GHz CPU and 32 GB RAM. The processing time for each step in ComMet's pipeline is shown in S6 Table.

### Addition of constraints for simulating metabolic states

To begin with, a steady-state flux space was defined by imposing a default set of constraints on all the reactions in the model. The bounds of all the reversible reactions were set to [-1,000 1,000] mmol gDW$^{-1}$ h$^{-1}$ and the irreversible reactions to [0 1, 000] mmol gDW$^{-1}$ h$^{-1}$. Same rules were applied to the 151 exchange reactions in the model, which were originally set to no uptake or efflux, depending on directionality. Following the definition of steady-state flux space, additional constraints were introduced to simulate two metabolic states: (a) Unconstrained substrate uptake (same as the model with default constraints) and (b) Constrained uptake of BCAAs. Setting both the upper and lower flux bounds of BCAA exchange reactions (HMR_9039: isoleucine uptake, HMR_9040: leucine uptake, HMR_9046: valine uptake) to zero resulted in the constrained model. The unconstrained condition serves as an ideal scenario where BCAAs and other nutrients are available for adipocytes.

### Preprocessing

Both the models were then preprocessed by removing all the blocked reactions (2,043 and 2,046 reactions in the unconstrained and constrained models respectively), which were the reactions incapable of carrying any flux under the imposed conditions. Using Flux Variability Analysis, the minimum and maximum steady-state flux ranges of the remaining reactions (4,067 and 4,064 reactions in the unconstrained and constrained models respectively) were then identified and subsequently used for analytical approximation of fluxes. After the preprocessing, the final range of variability of BCAA uptake reactions was still 0–1000 mmol gDW$^{-1}$ h$^{-1}$ in the unconstrained case, while both upper and lower bounds were 0 mmol gDW$^{-1}$ h$^{-1}$ in the constrained case. In addition, thermodynamic feasibility of the preprocessed models were checked. This check was performed using the *checkThermodynamicConsistency* function from the COBRA Toolbox [32] with the gurobi solver [33]. The reaction forming lipid droplet (HMR_obj) was set as the objective function in both models only for checking thermodynamic feasibility and this test did not reveal any unfeasible thermodynamic cycles.

### Analytical approximation of fluxes

A MATLAB implementation of the EP algorithm [19] that was available with the original publication was downloaded and installed. EP approximation was run with the following

parameters individually for each condition: (a) 1,000 as maximum number of iterations (b) $1e^{-5}$ as the precision accuracy and (c) $1e^8$ as beta. Each EP run resulted in statistics of marginal distributions (means and variances) for each reaction and a matrix describing covariances between all reaction flux distributions. The number of rows and columns in the resulting covariance matrix was equal to the number of reactions (4,067 and 4,064 in the unconstrained and constrained models respectively) and this matrix was used for the downstream PCA-based analysis.

## Differential flux statistics

The objective here was to obtain a Z-score ($Z_i^{flux}$) for each reaction, i, using means (E) and variances (Var) of flux distributions (v) in both conditions (denoted by subscripts 1 and 2 in Eq (1)). The Z-score was used to quantify the significance of the change in each flux distribution between considered conditions [34]. It was calculated as the difference between the means in each of the conditions divided by the square root of the sum of variances in the respective conditions.

$$Z_i^{flux} = \frac{E_2(v_i) - E_1(v_i)}{\sqrt{Var_2(v_i) + Var_1(v_i)}} \tag{1}$$

These Z-scores were then transformed into probabilities of change using a cumulative Gaussian distribution. These p-values represented the significance of change in fluxes between the conditions.

## Principal component analysis and basis rotation

The covariance matrix resulting from the analytical approximation of fluxes was used for the PCA-based decomposition of the flux spaces. Eigenvectors and eigenvalues of each covariance matrix were calculated. The variance explained by each vector was computed by normalising the eigenvalues. The number of vectors explaining 99.9% of the variance was then identified and all the non-zero loadings were rotated using varimax rotation. This step resulted in one matrix of reaction loadings for each condition, where the rows represented reactions and the columns represented PCs.

## Module extraction

The reactions whose loadings were within half of the maximal loading within each principal component were considered as part of the module (same as the criterion used in [20]). The number of reactions above the defined threshold varied in each component. The set of reactions present in at least one module in a condition, or the global modules, were identified and used for constructing the reaction networks shown in S3 Fig.

## Independent component analysis

For identifying BCAA-specific modules, the matrices containing reaction loadings from both conditions were concatenated. Next, a loading cut-off was calculated by summing the mean absolute deviation and median of the maximum absolute loadings. This cut-off was used to identify and remove reactions with low loadings in all the PCs.

To identify the optimum number of ICs, first, a bootstrapping analysis was performed using the FastICA algorithm implemented in icasso toolbox [35]. N ICs were computed in 100 iterations with random initial conditions, where N = [2 to 90, in steps of 1]. pow3 nonlinearity and symmetrical approach were used for the decomposition. Next, the consistency of the

estimated ICs across iterations was assessed by plotting a stability profile for each N using the BIODICA toolbox [36, 37]. S4(A) Fig shows that, on average, the stability of the clusters decreases with increasing N (grey lines). Next, 2-means clustering was used to group the stability measures into two lines: one with uniform stability distribution (blue line) and the other with low stability distribution (red line). The point of intersection of these two lines revealed the optimal number of IC as 20 (black vertical line).

Next, ICA was rerun with the identified optimum N (20) for 9000 iterations. To ensure reproducibility of the estimated features, a random number was explicitly set for each iteration. The resulting kurtosis values for all the ICs were examined and the features having kurtosis greater or equal to 1 and lesser or equal to -1 were extracted. The plot of estimation frequency (the number of runs in which each feature was estimated as an IC, S4(B) Fig) revealed that the top 20 features were estimated in about 70% of the iterations (S4(B) Fig). These were too low to extract modules describing meaningful biological differences (S5 Fig). Then, [38] was used to identify the knee point of this curve, indicated by the cyan vertical line in S4(B) Fig. All the features on the left of the cyan line were selected for further analysis. The selected features corresponded to 288 rotated-PCs that were distinct, and thus, described the metabolic differences between the two simulated conditions. Modules were then extracted from these rotated-PCs and the corresponding reactions were also visualised as a reaction network.

It is evident from Fig 4 that nearly the entire flux space can be recovered through ~500 principal components out of ~4000 in a given flux space. As the principal components removed prior to ICA-based comparison explain only about 0.1% of variation of flux space, they would not yield much information on the differences between flux spaces either. Retaining the components that are not significant determinants of metabolic behaviour could introduce noise in the subsequent analysis. ICA is the most time consuming step of the ComMet workflow (S6 Table). If the remaining components were included for ICA based comparison, it would also increase the columns in the input matrix for ICA (from about 1000 to over 8000) and thus invariably resulting in a tremendous increase in runtime.

## Network visualisation

To construct a reaction map or network from the global modules, two text files were generated: one graph file and an attributes file. The graph file described the connectivity between reactions and was built based on the connectivities defined in the metabolic network. Two reactions were defined to be connected if they shared either a product or a reactant. The following ubiquitous metabolites were removed from the calculation of connectivity—CoA, ubiquinol, ubiquinone, $NH_3$, $O_2$, $H_2O$, $H^+$, ATP, ADP, AMP, dADP, dATP, Pi, PPi, CTP, CDP, CMP, dCTP, dCDP, dCMP, UTP, UDP, UMP, dUTP, dUDP, dUMP, GTP, GDP, GMP, dGTP, dGDP, dGMP, ITP, IDP, IMP, dITP, dIDP, dIMP, TTP, TDP, TMP, dTTP, dTDP, dTMP, NADH, NADPH, $NAD^+$, $NADP^+$, $FADH_2$, FAD, $CO_2$, $Na^+$, $HCO_3^-$. The attributes file, on the other hand, described features of reactions. For each reaction, the following properties were identified: (1) the number of modules it is involved in, (2) list of all the modules involved, (3) subsystem to which it belongs and (4) its chemical equation. The reaction tables were written into semi-colon separated text files which were imported into Cytoscape for further investigation. In the case of the distinct modules network, the reaction tables from both the conditions were merged prior to import. These combined reactions were visualised as the distinct modules network in Cytoscape (Fig 5). In addition to the reactions shown in Fig 5, the network contained reactions from Extracellular Transport which were hidden from visualisation for the ease of interpretation. To construct metabolic maps, reactions from the distinct modules network belonging to subsystems of interest were extracted as submodels in MATLAB and

visualised in Cytoscape using the EFMviz workflow [39]. To visualise individual modules, reactions from modules of interest were selected in the distinct modules network (Fig 5) and extracted into a separate network in Cytoscape (Fig 8). All the network operations were automated through an R script (R v3.5.1) using the library, RCy3 [40]. NDEx [41] links for all the reaction networks have been provided for further interactive exploration.

## Supporting information

**S1 Fig. Summary of reaction flux statistics resulting from analytical approximation of fluxes.** (A) Histogram of reaction flux means from the unconstrained and (B) constrained simulations. (C) Histogram of reaction flux standard deviations from the unconstrained and (D) constrained simulations. Due to the very high number of reactions for the lowest values, their number of reactions have been indicated separately with arrows.
(TIF)

**S2 Fig. Flux distributions of 9 reactions among the top 30 reactions with most significant differences in flux statistics ($p < 0.05$) between the unconstrained and constrained simulations.** The reaction IDs and chemical equations have been shown above and below the plots respectively.
(TIF)

**S3 Fig. Reaction map or network of the global modules from both conditions.** The two networks represent the global modules from (A) unconstrained and (B) constrained adipocyte network. Nodes represent reactions and the edges indicate shared reactant/product. Node colour is mapped to the number of involved modules. The NDEx links https://bit.ly/globalModulesUncon and https://bit.ly/globalModulesCon can be used to study the networks interactively.
(TIF)

**S4 Fig. ICA optimization for selecting the number of features.** (A) Grey lines show the stability profiles of the bootstrapped Independent Component Analysis (ICA) runs. Blue and red dashed lines are results of two-line clustering with the optimal N determined as the point of their intersection (black vertical line). (B) Frequency of estimation of a feature as an independent component. Cyan vertical line marks the knee-point cut-off (288) for selecting distinct features.
(TIF)

**S5 Fig. Combined reaction map or network of modules extracted from the top 20 frequently occurring features.**
(TIF)

**S1 Table. Table with flux statistics of all the reactions from both simulations.**
(XLSX)

**S2 Table. Reactions with significant changes in flux statistics between unconstrained and constrained simulations.**
(XLSX)

**S3 Table. Subsystem distribution: Reactions from global modules—Unconstrained.**
(XLSX)

**S4 Table. Subsystem distribution: Reactions from global modules—Constrained.**
(XLSX)

**S5 Table. Table with reactions in the modules that were distinct between the simulations.**
(XLSX)

**S6 Table. Table with processing time for each step in ComMet's pipeline.**
(XLSX)

## Author Contributions

**Conceptualization:** Chaitra Sarathy, Marian Breuer, Martina Kutmon, Ilja C. W. Arts.

**Data curation:** Chaitra Sarathy.

**Formal analysis:** Chaitra Sarathy.

**Funding acquisition:** Ilja C. W. Arts.

**Methodology:** Chaitra Sarathy, Marian Breuer, Martina Kutmon, Michiel E. Adriaens.

**Project administration:** Chaitra Sarathy, Martina Kutmon, Ilja C. W. Arts.

**Software:** Chaitra Sarathy.

**Supervision:** Marian Breuer, Martina Kutmon, Chris T. Evelo, Ilja C. W. Arts.

**Validation:** Chaitra Sarathy.

**Visualization:** Chaitra Sarathy, Martina Kutmon.

**Writing – original draft:** Chaitra Sarathy.

**Writing – review & editing:** Chaitra Sarathy, Marian Breuer, Martina Kutmon, Michiel E. Adriaens, Chris T. Evelo, Ilja C. W. Arts.

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
