## [Decision Letter · Decision Letter 0]

29 Apr 2021

Dear Mrs Sarathy,

Thank you very much for submitting your manuscript "Comparison of metabolic states using genome-scale metabolic models" for consideration at PLOS Computational Biology.

As with all papers reviewed by the journal, your manuscript was reviewed by members of the editorial board and by several independent reviewers. In light of the reviews (below this email), we would like to invite the resubmission of a significantly-revised version that takes into account the reviewers' comments.

While the reviewers appreciated the attention to the research topic, they raised several substantial concerns about the manuscript based on which it cannot be accepted in its present form. In particular, reviewer #2 expressed concerns about the methodological and conceptual advance the study provides. However, we are willing to consider a substantially revised version in which all major issues raised (especially by reviewer #2) have been adequately addressed. It would be crucial i) to demonstrate that the predictions made by the ComMet workflow are different from what can be obtained by simpler approaches and ii) to provide some further validation of the predictions based on literature data. Please make sure that your revision addresses the specific points made by each reviewer.

We cannot make any decision about publication until we have seen the revised manuscript and your response to the reviewers' comments. Your revised manuscript is also likely to be sent to reviewers for further evaluation.

Sincerely,

Balazs Papp

Guest Editor

PLOS Computational Biology

Jason Papin

Editor-in-Chief

PLOS Computational Biology

While the reviewers appreciated the attention to the research topic, they raised several substantial concerns about the manuscript based on which it cannot be accepted in its present form. In particular, reviewer #2 expressed concerns about the methodological and conceptual advance the study provides. However, we are willing to consider a substantially revised version in which all major issues raised (especially by reviewer #2) have been adequately addressed. It would be crucial i) to demonstrate that the predictions made by the ComMet workflow are different from what can be obtained by simpler approaches and ii) to provide some further validation of the predictions based on literature data. Please make sure that your revision addresses the specific points made by each reviewer.

Reviewer's Responses to Questions

**Comments to the Authors:**

Reviewer #1: The review is uploaded as an attachment.

Reviewer #2: General

Authors extend the previous work done in this field by the Palsson group for the analysis of sampled flux distributions using PCA only that in this study they use distributions obtained from analytic approximation published recently. They also add an ICA piece to extract some additional modules that is marginally and incrementally different from the previous work in this area. They then show the workflow for the analysis of metabolism under two different cases, first when there are no constraints and then second, when there are constraints on the uptake of specific AA (Leucine, Valine and Isoleucine ). The work seems technically interesting but I am not sure this work has enough conceptual or methodological advance for PLoS CB and neither are the insights identified in this study.

Major Concerns

The paper is not structured well. It is written reasonably well but for example there are description of the methods in the results section (Lines 95-Line 120).

The major weakness is that the paper does not identify what specific aspects of the workflow is novel compared to Barrett paper that also applied PCA to the actual sampled distributions as opposed to the analytic distribution. A proper comparison would be to look at the differences and perhaps show that the PCA of the analytic piece is different and produces some new insights that was not obvious with the previous method of PCA on the sampled distributions as opposed to the approximate analytical distributions. If the advance is computational efficiency then authors should provide data supporting these claims. This reviewer seriously wonders if sampling based on ACHR is indeed a limitation at all.

The modules identified globally for the adipocyte network is then analysed but it is not clear whether the reactions constrained truly represent the physiology under obese/nonobese conditions before we can conclude that the changes in the flux space due to the lack of these constraints are meaningful. Furthermore the fact that one of the conditions is unconstrained space is not justified clearly.

If the aim of the paper is to demonstrate a workflow then it should be illustrated on a small toy network so that it is clear what the advantages of the workflow are. If the goal is to understand metabolism of adipocytes under disease conditions then more physiology constraints and arguments are needed with some validation based on data from literature.

Authors apply ICA to the filtered PCA results and the value of this step is not at all clear. First of can the authors compare these results with what happens when ICA is done on the entire space compared to filtered results. Why do we need the PCA step ? Also the reaction modules identified by ICA, how are they different from the subsystems annotation. Perhaps authors can compare these and show some additional modules. May be even compare with gene expression data as well to incorporate some of the work that the Palsson group has done around imodulons.

Finally the so called predictions of the ComMet workflow seems a bit trivial and easily predictable from a pure FBA simulation. Can the authors indicate why one would need to go through this entire exercise if we can use FBA or other sampling/coupling methods itself ?

**Have all data underlying the figures and results presented in the manuscript been provided?**

Reviewer #1: Yes

PLOS authors have the option to publish the peer review history of their article (what does this mean?). If published, this will include your full peer review and any attached files.

Reviewer #1: No

Reviewer #2: No

**Have the authors made all data and (if applicable) computational code underlying the findings in their manuscript fully available?**

Reviewer #2: None
---

## [Decision Letter · Decision Letter 1]

4 Oct 2021

Dear Sarathy,

We are pleased to inform you that your manuscript 'Comparison of metabolic states using genome-scale metabolic models' has been provisionally accepted for publication in PLOS Computational Biology.

Best regards,

Balazs Papp

Guest Editor

PLOS Computational Biology

Jason Papin

Editor-in-Chief

PLOS Computational Biology

Thank you for submitting a revised version of your manuscript. It has now been seen by one of the original reviewer and myself. We both think that the revision addressed all concerns and the manuscript has been significantly improved. Based on this, it is now acceptable for publication.

Reviewer's Responses to Questions

**Comments to the Authors:**

Reviewer #1: I have found the answers and the modifications to the paper satisfying.

**Have the authors made all data and (if applicable) computational code underlying the findings in their manuscript fully available?**

Reviewer #1: Yes

PLOS authors have the option to publish the peer review history of their article (what does this mean?). If published, this will include your full peer review and any attached files.

Reviewer #1: No

---

## [Editor Report · Acceptance letter]

2 Nov 2021

PCOMPBIOL-D-21-00382R1 

Comparison of metabolic states using genome-scale metabolic models

Dear Dr Sarathy,

I am pleased to inform you that your manuscript has been formally accepted for publication in PLOS Computational Biology. Your manuscript is now with our production department and you will be notified of the publication date in due course.

With kind regards,

Zsanett Szabo
